# Relationship between Child Care Exhaustion and Breastfeeding Type at Two and Six Months in a Cohort of 1210 Japanese Mothers

**DOI:** 10.3390/nu14061138

**Published:** 2022-03-08

**Authors:** Tomoya Suzuki, Keisuke Nojiri, Satoshi Higurashi, Yuta Tsujimori, Yasuhiro Toba, Kyoko Nomura

**Affiliations:** 1School of Medicine, Akita University, 1-1-1 Hondo, Akita 010-8543, Japan; tomo1502@gmail.com; 2Research and Development Department, Bean Stalk Snow Co., Ltd., Saitama 350-1165, Japan; keisuke-nojiri@beanstalksnow.co.jp (K.N.); s-higurashi@beanstalksnow.co.jp (S.H.); yuta-tsujimori@beanstalksnow.co.jp (Y.T.); y-toba@beanstalksnow.co.jp (Y.T.); 3Department of Environment Health Science and Public Health, Akita University Graduate School of Medicine, 1-1-1 Hondo, Akita 010-8543, Japan

**Keywords:** exclusive breastfeeding, childcare exhaustion, parenting stress, partial breastfeeding

## Abstract

This study investigated whether parenting stress is associated with breastfeeding type (exclusive or partial). Between 2014 and 2019, we recruited 1210 healthy mothers (mean age, 31.2 years; 65%, multiparity) from 73 obstetric institutions across all prefectures of Japan. Among these, 1120 mothers at two months and 1035 mothers at six months were investigated for parenting stress and breastfeeding type: exclusive versus otherwise (partial). Parenting stress was measured by a validated Japanese scale consisting of childcare exhaustion, worry about child’s development, and no partner support. Exclusive breastfeeding prevalence was 75% at two and 78% at six months. The total scores for childcare exhaustion and worry about child development were statistically higher in the partial breastfeeding group than in the exclusive breastfeeding group at two months but not at six months. A logistic regression model demonstrated that childcare exhaustion was significantly associated with an increased risk of having partial breastfeeding at two months after adjusting for the maternal Body Mass Index, parity, and baby’s current weight. However, the association was no longer significant at six months. The present study suggests that intervention for parenting stress at two months postpartum may promote prolonged exclusive breastfeeding.

## 1. Introduction

The World Health Organization (WHO) and United Nations International Children’s Emergency Fund (UNICEF) recommend that children initiate breastfeeding within the first hour of birth and be exclusively breastfed for the first six months of life [1]. Breastmilk is safe, clean, and contains antibodies that help protect against many common childhood illnesses [2]. Some studies have reported that exclusive breastfeeding helps protect against common infections during infancy and reduces the frequency and severity of infectious episodes [2] and sudden infant death syndrome [3].

The benefits are not only for infants but also for mothers. A study reported that longer breastfeeding duration was associated with a lower maternal risk of hypertension and cardiovascular disease, irrespective of the pregnancy Body Mass Index (BMI) and abdominal adiposity after delivery [4]. Furthermore, it was reported that breastfeeding is associated with a reduced risk of female reproductive cancers, such as endometrial [5], breast, and ovarian cancer [6]. Although the benefits of exclusive breastfeeding have been proven and about 90% of Japanese mothers agree with exclusive breastfeeding for the first six months, only 51.6% breastfeed their infant to any extent at one month after delivery [7]. Global analyses have also shown that, in most countries, the rates of exclusive breastfeeding are well below 50% at six months [8]. There are several reasons why it is difficult to maintain a high prevalence of breastfeeding. We previously reported that overweight or obese mothers, gestational week, cesarean section, nulliparity, older maternal age [9,10], and mental anxiety [11] were associated with an increased risk of not initiating/continuing breastfeeding.

In addition, another study reported that mothers’ concern about lactation/baby feeding is the most frequent reason for breastfeeding cessation during the first two months, reporting “My baby began to bite”, “My baby lost interest in nursing or began to wean him/herself”, or “Breast milk alone did not satisfy my baby” [11]. These statements reflect maternal frustrations about nursing, suggesting that their psychological distress may inhibit breastfeeding [12]. Indeed, one of the most serious psychological statuses, maternal depression, was previously considered an established risk factor for breastfeeding cessation [13,14].

In contrast, for healthy mothers, very few studies have investigated the impact of parenting distress on breastfeeding. Hence, in this study, we investigated parenting stress on the breastfeeding type, exclusive or partial breastfeeding, in healthy Japanese mothers without any underlying illnesses that require medication.

## 2. Materials and Methods

### 2.1. Study Design, Setting, and Participants

This study was cross-sectional. We recruited Japanese lactating women and their infants aged two and six months after delivery at 73 medical institutions, including 16 hospitals. The remaining sites were obstetrics clinics across all prefectures in Japan. The enrollment period spanned between October 2014 and May 2019. The details of our study were described elsewhere [15]. The inclusion criteria were as follows: (1) healthy singleton infants and (2) healthy mothers who were free from any underlying illness that required periodic hospital visits. In this regard, we defined “healthy mothers” as mothers who do not have underlying illnesses that require ongoing medication. Exclusion criteria were as follows: (1) hepatitis B-positive or hepatitis C-positive participants or participants with human immunodeficiency virus or human T-cell leukemia virus type 1 infections; (2) participants who were on medication for underlying illnesses; and (3) mothers, partners, or children who were not of Japanese ethnicity. Participants with illnesses mentioned above were excluded, because they were more likely to stop breastfeeding. We recruited 1210 mothers; however, 83 mothers did not provide valid responses, and five parents were not of Japanese ethnicity. Further excluding two mothers at two months and 87 mothers at six months who did not answer their breastfeeding type, and finally, 1120 mothers at two months and 1035 mothers at six months were included in the analyses (mean age, 31 years, Figure 1).

### 2.2. Definition of Breastfeeding Type

We collected information about breastfeeding at two and six months after delivery using a self-administered questionnaire by postal mail. Breastfeeding status was classified as follows: (1) breast milk only, (2) breast milk in combination with formula milk, or (3) formula milk only. Exclusive breastfeeding has been defined by the WHO [1] as “the infant has received only breast milk from his/her mother or a wet nurse, or expressed breast milk and no other liquids, or solids, except for drops or syrups consisting of vitamins, minerals, supplements, or medicines”. Thus, we defined 100% breast milk status as “exclusive” breastfeeding. We created a binary outcome variable of “exclusive” versus “otherwise”, namely “partial” breastfeeding that included the remaining status responses.

### 2.3. Questionnaire Survey for Health Information

The questionnaire included maternal information on sociodemographic factors (age, educational attainment, and employment status); body weight before and during pregnancy; height; child-rearing stress; and a clinical history of underlying diseases. Information concerning the pregnancy outcomes included parity (nulliparity or multiparity), gestational age at delivery, and delivery mode (cesarean section or otherwise, including vaginal delivery). Information concerning the infants included sex and body weight and height at birth, two months, and six months. Other family information included family structure, annual household income, and mothers’ employment status. Maternal child-rearing-related stress was assessed using a mother’s child care stress scale (CSS) [16], consisting of a checklist involving three subscales for mental and physical fatigue, worry over child-rearing, and lack of a husband’s support. The detailed items of the three subscales are presented in the Appendix A.

### 2.4. Data Analysis

We used CSS, because there were very few scales to measure parenting stress available to assess mothers’ perceptions of stress in child-rearing or translated into Japanese. The CSS scale was developed in Japanese and verified to have good reliability and validity [16]. The scale development was based on the psychological stress theory of Lazarus & Folkman [17], and the scale asks mothers how they perceive their parenting environment. By using the CSS to clarify mothers’ perceptions of stressful events associated with child-rearing, we were able to examine the mothers’ stress and how to intervene in the environment, including related situations and people. In particular, in Japan, there have been many cases of mothers who cannot cope with the stress of child-rearing. This inability leads to a loss of confidence about relating to their children and parents’ psychological distress. It may be associated with aggression in terms of the behavioral and emotional aspects (i.e., abuse) [18]. The original CSS consists of 33 items on a 9-subscale scale, which was challenging to use in a field study, because the follow-up rate may decrease. Instead, we used the short version of the CSS. In this study, we examined the reliability of the short version of the CSS by performing principal factor analyses with Varimax rotation. We confirmed that each item within each factor had high loading (see the Appendix A). Three domains consisted of childcare exhaustion (Cronbach’s alpha was 0.836 for two months and 0.853 for six months), worry about child development (Cronbach’s alpha was 0.863 for two months and 0.880 for six months), and no support from a partner (Cronbach’s alpha was 0.763 for two months and 0.828 for six months). The factor loadings ranged from 0.547 to 0.878. After confirming the validity and reliability of the CSS, we used a *t*-test to investigate the mean differences between exclusive breastfeeding and partial breastfeeding. A logistic regression model was used to identify the factors associated with an increased risk of not initiating (at two months) or continuing (at six months) exclusive breastfeeding. To adjust for covariates, multivariable logistic regression analyses were performed by entering a significant variable (*p* < 0.20 based on the chi-square test) in univariate models. If more than one CSS domain was selected in the univariate model, a multivariate model was individually built for each domain. Then, we estimated the odds ratios and the 95% confidence interval (CI). The significance level was set at 5% (two-tailed), and all analyses were conducted using STATA (Version 17; StataCorp, College Station, TX, USA).

## 3. Results

Table 1 shows the participants’ baseline characteristics. The majority of mothers had a vaginal delivery (88%), multiparity (65%), established exclusive breastfeeding (*n* = 835, 75%) at two months, and had exclusive breastfeeding (*n* = 803, 78%) at six months. Most mothers graduated from high school or college and were not in labor during the investigation phase. The median age of the infants was 59 days, with an interquartile range of 38–66 days at two months and 176 days with an interquartile range of 154–192 days at six months. The boy (*n* = 602) and girl (*n* = 518) median heights and weights were 56.0 cm and 5700 g and 54.0 cm and 4462 g, respectively, at two months, and 66.0 cm and 7740 g and 64.1 cm and 7100 g, respectively, at six months. These values were similar to those reported by the National Growth Survey on Newborn Infants in 2010 [7].

Table 2 shows the mean ± standard deviation of the three domains of the Mothers’ Child Care Stress Scale at two and six months, according to breastfeeding type. At two months, the means of the total scores for child care exhaustion and worry about child development were higher in partial breastfeeding mothers than in exclusive breastfeeding mothers (*p* < 0.001 and *p* = 0.002, respectively). At six months, the total scores for child care exhaustion and worry about child development decreased compared to those at two months. However, there were no significant differences between the exclusive and partial breastfeeding items. For no partner support, the total score was unchanged at two and six months, and there was no significant difference in the mean scores of any items between exclusive and partial breastfeeding at two and six months.

Table 3 shows the factors associated with exclusive breastfeeding failure at two months. Logistic regression models demonstrated that two domains of the CSS, including childcare exhaustion and worry about child development, were associated with an increased risk of partial breastfeeding (both *p* < 0.001). Significant variables in univariate logistic models selected at *p* < 0.2 included child care exhaustion, worry about child development, maternal BMI, the number of delivery experience, annual income, birth weight, and current infant weight. In Model 1 (Figure 2), where childcare exhaustion was included, the risk of partial breastfeeding increased with childcare exhaustion (OR 1.23 per 5-point increase; 95% CI 1.07–1.42), obesity (OR 1.97, 95% CI: 1.23–3.16) compared to normal weight, and one-time delivery experience (OR 1.55, 95% CI: 1.13–2.12), while the risk decreased with a 1000-g increase of current infant weight (OR 0.71, 95% CI: 0.58–0.87). In Model 2, where worry about child development was included, the risk of partial breastfeeding increased with obesity compared to normal weight (OR 1.96, 95% CI: 1.23–3.12), one-time delivery experience (OR 1.45, 95%CI: 1.05–2.00), and a 1000-g increase of current infant weight (OR 0.69, 95% CI: 0.56–0.85).

Table 4 shows the factors associated with the risk of not having exclusive breastfeeding at six months. Significant variables in the univariate logistic regression models selected at *p* < 0.2 included worry about child development, maternal BMI, delivery method, parity, annual income, mother’s education, employment status, birth weight, and current infant weight. Adjusting for these variables, a multivariate logistic model demonstrated that one-time delivery experience (OR 1.77, 95% CI: 1.22–2.57) and the lowest quartile of annual income (OR 1.83, 95% CI: 1.07–3.12) were associated with an increased risk of partial breastfeeding. In contrast, current infant weight was associated with a decreased risk (OR 0.62 per 1000-g increase, 95% CI: 0.49–0.77). Worry about child development did not become significant.

## 4. Discussion

This study investigated the relationship between parental stress and breastfeeding type among healthy Japanese mothers. The percentage of exclusive breastfeeding exceeded 70%, higher than the Japanese average of 51% [7]. The high prevalence of exclusive breastfeeding in our study may be due to their characteristics of being conscious about breastfeeding in addition to their health status [19]. We demonstrated that, among the three domains of parental stress, “childcare exhaustion” was associated with an increased risk of partial breastfeeding at two months; however, this association disappeared at six months.

The findings suggested that maternal stress at two months may play a key role in breastfeeding establishment and its continuation. Previous research also reported that parenting stress was shown to have an effect on maternal postpartum depression starting from the child’s third month [20]. Intervention after birth is currently available as a checkup one month after birth at obstetric institutions; however, these do not usually include a parental stress assessment. Alternatively, in Japan, a handbook of mothers and children, which was firstly distributed in Japan in 1946 to every expectant mother for the health promotion of mothers and children before and after birth, does not include such an assessment [21]. Instead, it simply includes a few questions such as “Are there any people who can help you with child-raising” and “Do you have any difficulties or worries about parenting?”. The study results suggested that the early phase of one to two months postpartum may be an essential period for the initiation of exclusive breastfeeding. In addition, one study underscored the importance of adherence to a healthy diet as a coping factor for stress [22]. Hence, the assessment of mental health status, along with lifestyle, may need to be included at one month after birth, and the handbook of mothers and children also needs to be updated [21]. Mental health screening, including parental stress, is also essential for detecting postpartum depression. Previous studies have demonstrated that parenting stressors have been associated with a greater risk of depressive symptoms, including baby feeding and sleeping problems, excessive crying and illness, affordability of childcare services, and balancing work–family demands [23,24]. A study showed that depressive symptomatology during the postpartum period negatively influenced breastfeeding practices [25]. A systematic review and meta-analysis based on 30 studies suggested that maternal depression during pregnancy is associated with decreased breastfeeding initiation [26]. Hence, the reduction of parental stress in the early phase after delivery is an important area of research in future studies.

We found that the total scores for childcare exhaustion and worry about child development were higher at two months than at six months. The higher stress scores at two months may be natural for every single mother adjusting to the intensive commitment to newborn care and worry about their child’s development. If a third person existed to help physically and psychologically in the early phase after birth, the scores of the two dimensions may decrease to some extent. A third person may be a partner or grandparents, in most cases; however, support from partners was not associated with breastfeeding practice in our study. A U.S. research study reported that living with grandparents was associated with low odds of breastfeeding for six months [27,28]; however, another previous study identified spousal supportiveness as a critical resource to help reduce maternal parenting stress [29]. Our study found that no relationship existed with partner support, which may be due to the scale used. We measured partner support by four sentences: “He does not understand my suffering…”, “He mainly puts his life first...”, “He is not supportive…“, and “His support is not always helpful…”. These four conditions reflect the human relationship between a mother and father. If the parents had a difficult relationship, the wife would report higher values for these items, even if the husband was deeply committed to parenting. Hence, the interpretation of “no partner support” may require careful attention.

Nevertheless, a male partner’s time commitment or actual involvement in child-rearing is still not common in Japan, because the gender division of labor is strongly embedded in the mindset of Japanese people. In the White paper on gender equality in 2019 [30], more than 50% of both men and women still agree that men are breadwinners while women take care of the family in Japan. Additionally, as the birthrate is declining, family homes are one-generational and live apart from their grandparents. The so-called nuclear family makes it difficult for women to balance work and family responsibilities, because a mother has little chance to receive childcare support from their parents.

In our study, obesity, delivery experience, income and educational attainment of mothers, and current birth weight, in addition to parenting stress, were associated with an increased risk of partial breastfeeding at two months. Previously, these factors were accumulated as evidence reported as the socioeconomic factors of income and educational attainment. In addition, maternal obesity is an independent and important factor of exclusive breastfeeding, as we have already noted from a teaching hospital in Japan [9] and a meta-analysis about maternal overweight/obesity [10]. We previously reported that primiparous mothers of late childbearing ages (35 years or older) were at the most significant risk of not initiating exclusive breastfeeding [31]. Our study did not find an effect of older age on exclusive breastfeeding establishment. The age discrepancy for older mothers between the present and previous study [31] may be due to the distribution of the population in question, as the mean age of the previous population was 35 years and older. In contrast, the mean age of this study was 31 years.

## 5. Limitations

This study had several limitations that need to be addressed. First, our participants were primarily recruited from clinics, indicating that they were healthy women without complications at the time of delivery. Furthermore, more than half of our participants were multiparous women who were more likely to breastfeed [20], leading to a high prevalence of breastfeeding practices. Second, the present study was based on a cross-sectional investigation. Thus, a causal relationship could not be determined between exclusive breastfeeding and maternal parenting stress. Third, our study’s outcome of breastfeeding type was based on self-reporting. Although we included both exclusively as the outcomes of interest, self-ascertainment might be prone to misclassification bias. Fourth, although our study population consisted of healthy mothers that did not require periodic hospital visits and medication, mild levels of underlying diseases, including mental illness or mental health issues during the pre-pregnancy period, might influence breastfeeding practices. In addition, we did not measure depressive symptoms that may influence mothers’ perceptions of parenting stress. Hence, our results require careful interpretation.

## 6. Conclusions

This study of healthy mothers in Japan demonstrated that the parental stress of “childcare exhaustion” was significantly associated with an increased risk of partial breastfeeding at two months, even after adjusting for related factors. However, mothers’ parenting stress was no longer significant at six months, suggesting the importance of early intervention against the maternal psychological burden on parenting at two months after delivery so that mothers may be able to continue exclusive breastfeeding.

## Figures and Tables

**Figure 1 nutrients-14-01138-f001:**
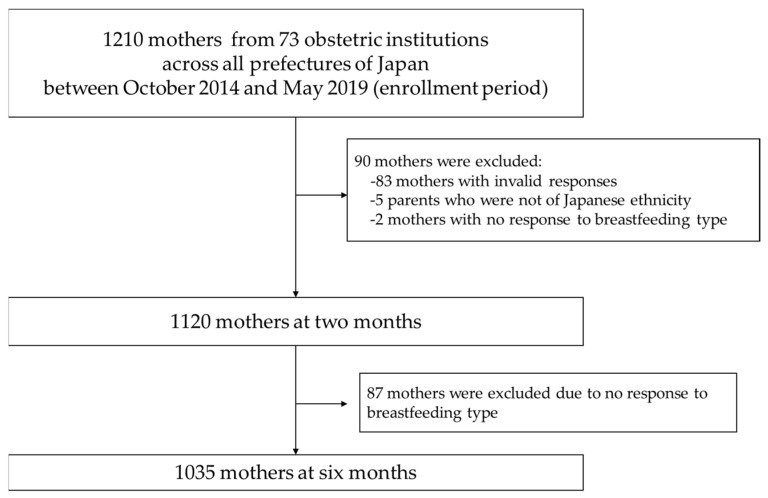
Study enrollment flowchart.

**Figure 2 nutrients-14-01138-f002:**
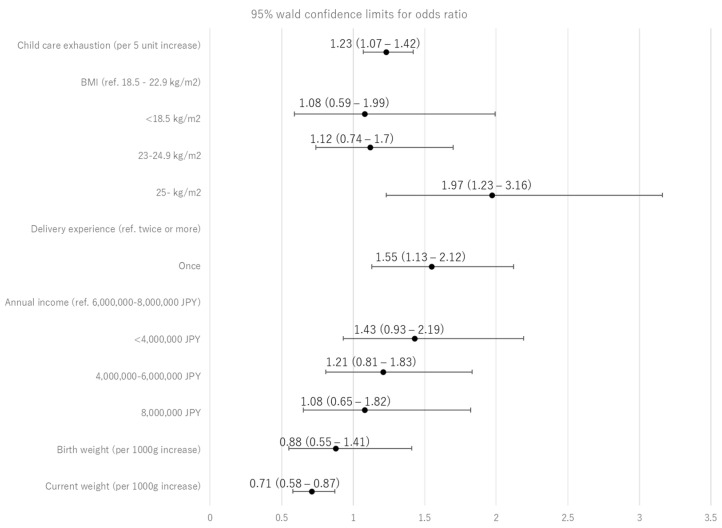
Adjusted odds ratios of partial breastfeeding at two months (Model 1: childcare exhaustion model).

**Table 1 nutrients-14-01138-t001:** Baseline characteristics.

		Two Months (*n* = 1120)				*p*	Six Months (*n* = 1035)				*p*
		Exclusive		Partial			Exclusive		Partial		
		(*n* = 835, 75%)		(*n* = 285, 25%)			(*n* = 803, 78%)		(*n* = 232, 22%)		
		N	%	N	%		N	%	N	%	
Mother											
Age, years						0.243					0.574
	−30	373	45	116	41		316	40	96	42	
	31−	462	55	169	59		484	60	135	58	
BMI, kg/m^2^						0.101					0.112
	<18.5	56	7	21	7		109	14	36	16	
	18.5–22.9	555	67	175	62		535	68	136	60	
	23–24.9	141	17	46	16		88	11	28	12	
	25−	80	10	42	15		58	7	26	12	
Delivery method						0.396					0.074
	Cesarean section	95	11	38	13		87	11	35	15	
	Vaginal delivery	735	89	247	87		712	89	195	85	
Delivery experience						0.002					0.082
	One time	265	32	119	42		270	34	92	40	
	Twice or more	566	68	165	58		529	66	138	60	
Socioeconomic factors											
Annual income (JPY)						0.196					0.020
	<4,000,000	202	25	85	31		183	24	72	33	
	4,000,000–6,000,000	284	35	98	36		279	36	79	36	
	6,000,000–8,000,000	197	25	57	21		189	24	48	22	
	8,000,000<	119	15	34	12		122	16	22	10	
Mother’s educational attainment						0.397					0.007
	Junior high school/ High school/Others	181	22	66	23		160	20	61	27	
	Some college/ technical	343	41	126	44		332	41	104	45	
	4-y college/graduate degree	306	37	92	32		309	39	64	28	

Mother’s employment status						0.419					0.151
	Employed	368	44	133	47		344	43	110	48	
	Unemployed	464	56	150	53		458	57	118	52	
Infant, median, 25–75%											
Boys											
	Birth height, cm	49.6	48.4–50.6	49.6	48.0–50.8	0.893	-	-	-	-	
	Birth weight, g	3108	2882–3354	3120	2865–3344	0.533	-	-	-	-	
	Current height, cm	55.0	53.2–57.0	54.5	53.3–57.0	0.584	66.0	64.0–68.0	65.1	63.0–67.3	0.107
	Current weight, g	4900	4385–5620	4588	4144–5340	0.008	7805	7121–8400	7500	6983–7900	0.001
Girls											
	Birth height, cm	49.2	48.0–50.0	49.0	48.0–50.0	0.516	-	-	-	-	
	Birth weight, g	3028	2806–3272	2970	2756–3174	0.067	-	-	-	-	
	Current height, cm	54.0	52.5–56.0	53.3	52.0–55.1	0.023	64.4	62.2–66.0	63.5	61.6–65.0	0.048
	Current weight, g	4525	4101–5290	4262	3824–4880	<0.001	7200	6615–7840	6945	6300–7400	0.002

Note If the summation of each category variable does not reach the total numbers of breastfeeding types, it indicates missing values in the category. The *p*-value was calculated by a chi-square or *t*-test, depending on the type of each variable (i.e., a categorical or continuous variable).

**Table 2 nutrients-14-01138-t002:** Means of three domains of the Mothers’ Child Care Stress Scale according to the breastfeeding type at two and six months.

	Two Months				*p*	Six months				*p*
	Exclusive Breastfeeding		Partial Breastfeeding			Exclusive Breastfeeding		Partial Breastfeeding		
	Mean	*SD*	Mean	*SD*		Mean	*SD*	Mean	*SD*	
Child care exhaustion	16.62	5.44	18.11	5.38	<0.001	15.26	5.79	14.87	5.00	0.378
Worry about child’s development	9.35	3.75	10.26	4.28	0.002	8.54	3.72	9.07	4.07	0.076
No support from partner	7.62	3.74	7.47	3.62	0.547	7.85	4.07	7.5	3.85	0.244

The *p*-value was calculated by a *t*-test. SD indicates standard deviation.

**Table 3 nutrients-14-01138-t003:** Factors associated with partial breastfeeding at two months.

			Univariate Model			Multivariate Model					
						Model 1			Model 2		
						(*n*= 968, Exhaustion)			(*n* = 968, Worry)		
			OR	95% CI		OR	95% CI		OR	95% CI	
				Lower	Upper		Lower	Upper		Lower	Upper
Mothers’ Child care Stress Scale *											
	Child care exhaustion		1.29	1.14	1.47	1.23	1.07	1.42	-	-	-
	Worry about child development		1.33	1.13	1.57	-	-	-	1.20	0.99	1.45
	No support from partner		0.95	0.79	1.14	-	-	-	-	-	-
Mother											
	Age, years					-	-	-	-	-	-
		31−	1.00	-	-						
		−30	1.18	0.90	1.55						
	BMI, kg/m^2^										
		<18.5	1.19	0.70	2.02	1.08	0.59	1.99	1.12	0.61	2.07
		18.5–22.9	1.00	-	-	1.00	-	-	1.00	-	-
		23–24.9	1.04	0.71	1.50	1.12	0.74	1.70	1.11	0.74	1.67
		25−	1.67	1.11	2.51	1.97	1.23	3.16	1.96	1.23	3.12
Delivery method											
		Cesarean section	1.19	0.80	1.78						
		Vaginal delivery	1.00	-	-						
Delivery experience											
		One time	1.54	1.17	2.03	1.55	1.13	2.12	1.45	1.05	2
		Twice or more	1.00	-	-	1.00	-	-	1.00	-	-
Socioeconomic factors											
Annual income (JPY)											
		<4,000,000	1.45	0.99	2.15	1.43	0.93	2.19	1.40	0.92	2.15
		4,000,000–6,000,000	1.19	0.82	1.73	1.21	0.81	1.83	1.22	0.81	1.83
		6,000,000–8,000,000	1.00	-	-	1.00	-	-	1.00	-	-
		8,000,000<	0.99	0.61	1.6	1.08	0.65	1.82	1.08	0.65	1.81
Mother’s educational attainment						-	-	-	-	-	-
	Junior high school/High school/Others		1.21	0.84	1.75						
	Some college/technical		1.22	0.90	1.67						
	4-y college/graduate degree		1.00	-	-						
Mother’s employment status						-	-	-	-	-	-
		Employed	0.89	0.68	1.17						
		Unemployed	1.00	-	-						
Infant											
	Birth weight **		0.73	0.50	1.07	0.88	0.55	1.41	0.93	0.58	1.48
	Current weight **		0.68	0.57	0.81	0.71	0.58	0.87	0.69	0.56	0.85

* Indicates a risk ratio per 5-unit increase. ** Indicates a risk ratio per 1000-g increase.

**Table 4 nutrients-14-01138-t004:** Factors associated with partial breastfeeding at six months.

			Univariate Model				Multivariate Model (*n* = 741)		
			OR	95% CI		*p*	OR	95% CI	
				Lower	Upper			Lower	Upper
Mothers’ Child Care Stress Scale *									
	Child care exhaustion		0.94	0.83	1.07	0.378			
	Worry about child development		1.19	0.99	1.43	0.063	1.06	0.83	1.35
	No support from partner		0.90	0.74	1.08	0.244			
Mother									
	Age					0.574			
		−30	1.00	-	-				
		31−	0.92	0.68	1.24				
	BMI					0.115			
		<18.5	1.30	0.85	1.98		1.20	0.70	2.04
		18.5–22.9	1.00	-	-		1.00	-	-
		23–24.9	1.25	0.79	1.99		1.51	0.88	2.59
		25–	1.76	1.07	2.91		1.48	0.77	2.83
Delivery method						0.075			
		Cesarean section	1.47	0.96	2.24		1.58	0.94	2.64
		Vaginal delivery	1.00	-	-		1.00	-	-
Delivery experience						0.083			
		One time	1.31	0.97	1.77		1.77	1.22	2.57
		Twice or more	1.00	-	-		1.00	-	-
Socioeconomic factors									
Annual income (JPY)						0.021			
		<4,000,000	1.55	1.02	2.35		1.83	1.07	3.12
		4,000,000–6,000,000	1.12	0.75	1.67		1.14	0.69	1.88
		6,000,000–8,000,000	1.00	-	-		1.00	-	-
		8,000,000<	0.71	0.41	1.24		0.84	0.44	1.60
Mother’s educational attainment						0.008			
		Junior high school/High school/Others	1.84	1.24	2.74		1.66	0.99	2.75
		Some college/technical	1.51	1.07	2.14		1.19	0.78	1.82
		4-y college/graduate degree	1.00	-	-		1.00	-	-
Mother’s employment status						0.151			
		Employed	1.00	-	-		1.00	-	-
		Unemployed	0.81	0.60	1.08		0.76	0.52	1.11
Infant									
	Birth weight **		0.72	0.47	1.09	0.117	1.08	0.62	1.87
	Current weight **		0.67	0.56	0.81	<0.001	0.62	0.49	0.77

* Indicates a risk ratio per 5-unit increase. ** Indicates a risk ratio per 1000-g increase.

## Data Availability

The data presented in this study are available on request from the corresponding author.

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
