# Peer review of "Relationship between Child Care Exhaustion and Breastfeeding Type at Two and Six Months in a Cohort of 1210 Japanese Mothers"

_nutrients, 2022, doi:10.3390/nu14061138_

Round 1

Reviewer 1 Report

The article by Suzuki et al. was very interesting to read, they explore the role of psychological stress in the promotion of exclusive breastfeeding, which is essential and is increasingly being considered in policy interventions.

In my opinion, it is very well described and organized and I would like to give some comments to help the authors to improve the article:

  • The introduction is very well described. I would simply like to mention some quotes in case they could help to argue the texts both in the introduction and in the discussion. These articles highlight the maternal psychological component in relation to breastfeeding adherence: PMID: 34682201; PMID: 33007816.
  • In section 2.1. I would include the flow diagram as a figure. Regarding the hypothesis that race influences (line 84), I fully agree, however, in this article this factor is not going to be explored in the models.
  • Section 2.4. was vaginal delivery considered? are the neonatal anthropometric data at discharge or at birth?
  • The CSS scale has already been validated in other contexts (reference 17), why was it re-validated in the Japanese context? there are no data in Japanese population? have the items been modified? On the other hand, how the scores of each dimension were extracted should be described in the text.
  • Regarding the results, I would like to congratulate you for the high exclusive breastfeeding rates achieved (line 139) indicating good health promotion policies. However, if you could give the neonatal age in months instead of days it would be more informative.
  • Table 1. Could comparisons be made at two and six months, between exclusive and partial BM?
  • Table 2. I think what is more informative is the total of the dimensions and not the score of each item within each dimension, which should be eliminated.
  • As for the discussion, I wanted to congratulate you on it and reiterate the quotes from the above in case they might help you.
  • As for the limitations, I do not consider them to have a small sample size (they recruited over 2000 women), also, in the fourth limitation, they mention self-registration as a limitation, which although I agree that it may be a bias, with the population size and the model fit variables I think it may be diluted.
  • As for the conclusion, it is very accurate. However, the title should be modified to have a greater impact. For example, "Japanese Human Milk Study" and "Prospective Cohort Study of" could be deleted, but "child care exhaustion as a predictor of breastfeeding type" or something similar should be included.

Minor comments:

  • Check the author's guidelines, there are typographical errors throughout the text, as well as different fonts.
  • In the tables, it is necessary to put a table footnote explaining the descriptive and statistical analysis applied.
  • Combine sections 2.1. and 2.2.

Author Response

Reviewer 1

We thank the Reviewer #1 for the valuable comments which we believe would improve the quality of our manuscript. We respond to the comments from the reviewer 1 below one by one.

#1 The introduction is very well described. I would simply like to mention some quotes in case they could help to argue the texts both in the introduction and in the discussion. These articles highlight the maternal psychological component in relation to breastfeeding adherence: PMID: 34682201; PMID: 33007816.

→We thank the reviewer for the productive suggestion and accordingly included these two references in the Introduction in the revised manuscript.

Ref

Gila-Díaz A, Carrillo GH, López de Pablo ÁL, Arribas SM, Ramiro-Cortijo D. Association between Maternal Postpartum Depression, Stress, Optimism, and Breastfeeding Pattern in the First Six Months. Int J Environ Res Public Health. 2020 Sep 30;17(19):7153. doi: 10.3390/ijerph17197153.

→We included this study in the Introduction and replace reference [15] with this study.

Line 61-62

Indeed, one of the most serious psychological statuses, maternal depression, was pre-viously considered an established risk factor for breastfeeding cessation [14,15].

Gila-Díaz A, Herranz Carrillo G, Arribas SM, Ramiro-Cortijo D. Healthy Habits and Emotional Balance in Women during the Postpartum Period: Differences between Term and Preterm Delivery.Children (Basel). 2021 Oct 18;8(10):937. Doi: 10.3390/children8100937.

-this study was included as reference [23] in Discussion as follows.

Line 239-240,

In addition, one study underscored the importance of adherence to a healthy diet as coping factor for stress [23].

#2 In section 2.1. I would include the flow diagram as a figure. Regarding the hypothesis that race influences (line 84), I fully agree, however, in this article this factor is not going to be explored in the models.

→As the reviewer suggested, we included a figure of enrollment flowchart (Figure 1). We agree with the reviewer that since our target population is a homogeneous ethnic group, we were unable to estimate the impact of race on breastfeeding practice. Accordingly, we deleted the following sentence.

Line 82-83 in previous version of manuscript

We limited participation to people of Japanese ethnicity because we hypothesized that race is a fundamental determinant of breastfeeding practices.

#3 Section 2.4. was vaginal delivery considered? Are the neonatal anthropometric data at discharge or at birth?

→Yes, we measured “vaginal delivery” along with “cesarean section”. In our analyses, we created a binary variable of “delivery methods (cesarean section or otherwise).” The neonatal anthropometric data were obtained at birth. We included these informations as follows.

Line 97-103

2.3. Questionnaire survey for health information

The questionnaire included maternal information on socio-demographic factors (age, educational attainment, and employment status), body weight before and during pregnancy, height, child-rearing stress, and a clinical history of underlying diseases. Information concerning pregnancy outcomes included parity (nulliparity or multiparity), gestational age at delivery, and delivery mode (cesarean section or otherwise including vaginal delivery). Information concerning infants included sex, and body weight and height at birth, two months and six months.

#4 The CSS scale has already been validated in other contexts (reference 17), why was it re-validated in the Japanese context? There are no data in Japanese population? Have the items been modified? On the other hand, how the scores of each dimension were extracted should be described in the text.

→When we conceived the study design of Japanese Human Milk Study, we reviewed various scales of parenting stress but there were very few scales available to assess mothers’ perception of stress in child rearing nor translated in Japanese. The CSS scale is developed with Japanese and verified good reliability and validity. The scale development was based on the psychological stress theory of Lazarus & Folkman (Stress, Appraisal, and Coping, 1984), and the scale asks mothers how they perceive their parenting environment. By using CSS to clarify mothers’ perceptions of stressful events associated with child rearing, we were able to examine not only mothers’ stress, but also how to intervene in the environment, including related situations and people. In particular, in Japan, there have been many cases of mothers who are unable to cope with the stress of child rearing, which leads to loss of confidence of how to relate to their children, parents' psychological distress, and may be associated with aggression on behavioral and emotional aspects (i.e., abuse). The original CSS consists of 33 items on a 9-subscale scale, which was difficult to use in a field study, because the follow-up rate may decrease. Instead, we used the short version of the CSS, which has been tested for reliability and conceptual validity as you see in reference #17. In this study, we examined the reliability of the short version of the CSS by performing principal factor analyses with varimax rotation and confirmed that each item within each factor had high loading which were shown in appendix. As the reviewer suggested, we added the following process how the scores of each dimension were extracted in Method section.

Line 109-123,

2.4. Data analysis

We used CSS because there were very few scales to measure parenting stress available to assess mothers’ perception of stress in child-rearing nor translated in Japanese. The CSS scale was developed with Japanese and verified to have good reliability and validity [17]. The scale development was based on the psychological stress theory of Lazarus & Folkman [18], and the scale asks mothers how they perceive their parenting environment. By using CSS to clarify mothers’ perceptions of stressful events associated with child-rearing, we were able to examine mothers’ stress and how to intervene in the environment, including related situations and people. In particular, in Japan, there have been many cases of mothers who cannot cope with the stress of child-rearing. This inability leads to a loss of confidence about relating to their children and parents' psychological distress. It may be associated with aggression on behavioral and emotional aspects (i.e., abuse) [19]. The original CSS consists of 33 items on a 9-subscale scale, which was challenging to use in a field study because the follow-up rate may decrease. Instead, we used the short version of the CSS. In this study, we examined the reliability of the short version of the CSS by performing principal factor analyses with Varimax rotation. We confirmed that each item within each factor had high loading (see Appendix).

#5 Regarding the results, I would like to congratulate you for the high exclusive breastfeeding rates achieved (line 139) indicating good health promotion policies. However, if you could give the neonatal age in months instead of days it would be more informative.

→We thank the reviewer. We presented exclusive breastfeeding rates in month as follows.

Line, P146-152

  1. Results

Table 1 shows participants’ baseline characteristics. The majority of mothers had a vaginal delivery (88%), multiparity (65%), established exclusive breastfeeding (n=835, 75%) at two months, and had exclusive breastfeeding (n=803, 78%) at six months. Most mothers graduated from high school or college and were not in labor during the investigation phase. The median age of the infants was 59 days, with an interquartile range of 38-66 days at two months and 176 days with an interquartile range of 154-192 days at six months.

#6 Table 1. Could comparisons be made at two and six months, between exclusive and partial BM?

→We included p-values between exclusive and partial breastfeeding at two and six months. Please refer to Table 1.

#7 Table 2. I think what is more informative is the total of the dimensions and not the score of each item within each dimension, which should be eliminated.

→Referring the reviewer’s suggestion, we have updated Table 2 (Please see).

#8 As for the limitations, I do not consider them to have a small sample size (they recruited over 2000 women), also, in the fourth limitation, they mention self-registration as a limitation, which although I agree that it may be a bias, with the population size and the model fit variables I think it may be diluted.

We thank the reviewer for such generous comments. Accordingly, we deleted the first limitation but remained the fourth limitation because the dilution does not cancel bias.

Line 292-306

  1. Limitations

This study had several limitations that need to be addressed. First, our participants were primarily recruited from clinics, indicating that they were healthy women without complications at the time of delivery. Furthermore, more than half of our participants were multiparous women who were more likely to breastfeed [20], leading to a high prevalence of breastfeeding practices. Second, the present study is based on a cross-sectional investigation. Thus, a causal relationship cannot be determined between exclusive breastfeeding and maternal parenting stress. Third, our study’s outcome of breastfeeding type was based on self-reporting. Although we included both exclusive as the outcome of interest, self-ascertainment might be prone to misclassification bias. Fourth, although our study population consisted of healthy mothers that did not require periodic hospital visits and medication, mild levels of mental illness or mental health issues during the pre-pregnancy period may not be ruled out. In addition, we did not measure depressive symptoms that may influence mothers’ perception of parenting stress. Hence, our results require careful interpretation.

#9 For example, "Japanese Human Milk Study" and "Prospective Cohort Study of" could be deleted, but "child care exhaustion as a predictor of breastfeeding type" or something similar should be included.

Referring to reviewer’s suggestion, we changed the title to “Relationship between child care exhaustion and breastfeeding type at two and six months in a cohort of 1,210 Japanese mothers”

#10 Check the author's guidelines, there are typographical errors throughout the text, as well as different fonts. In the tables, it is necessary to put a table footnote explaining the descriptive and statistical analysis applied.

We included a table footnote explaining statistical method under Table 1 and 2. We excluded p-value in Table 3 because the revised Table 1 included p-value referring to reviewer’s suggestion.

Combine sections 2.1. and 2.2.

→combined.

Reviewer 2 Report

Overall: This study sought to evaluate if parenting stress in healthy Japanese mothers is associated with breastfeeding type. Strengths include recruitment from 73 obstetric institutions and across all several states/prefectures which resulted in a generous sample size. This is a well-conducted study and well written manuscript. The authors do a good job of noting strengths and limitations of their research. One key recommendation I have is for authors to clarify what is meant by “healthy females”. Specifically, I am curious if any of the females are known to have depression, anxiety, or other mental health diagnoses. Mild to moderate mental health may be treated without medications and the same can be true for several other chronic disease (e.g., prediabetes may be treated initially through lifestyle change). I am assuming that a female not being on medication is a proxy for having no chronic medical problems but this should be explicit. In addition, while the findings are interesting and I agree with the suggested recommendation that parenting stress should be evaluated, previous data shows high correlation between depression, anxiety and parenting stress. These may be confounding factors which does not appear to have been measured. Below are some specific comments by section:

Abstract

  • No specific concerns

Introduction

  • I appreciate the review of benefits of breastfeeding for child and mother in the introduction. The authors cite some reasons in Line 51-55 on the reasons why some mothers do not exclusively breast feed. While data does support these demographic variables associated with unexclusive breastfeeding, there are other socioeconomic reasons along with the widespread availability of formula that should be considered being mentioned based on applicability in their population.
  • As noted above, it is not clear what is being defined as a healthy mother. Did the authors confirm that mothers did not have any chronic medical diagnoses? What about medical diagnoses that may have milder forms that do not require medication?

Methods

  • Where the rates of maternal depression/anxiety evaluated at all? As noted above, this has been shown to be independently associated with higher parenting stress
  • I think it is fine to limit to Japanese population but I do not know if I would agree with the phrase “race is a fundamental determinant of breastfeeding practices”. I would justify or modify the sentence
  • Line 91 – What is the meaning of “by post”? This is unclear.
  • Line 107 – the sentence is written in future tense while the remaining methods is written in past tense. Would maintain consistency
  • It would be helpful to have some brief information about the scale that was used to measure stress. It is not a well-known scale and there is limited data from what I see from a quick literature search

Results

  • Well written and described

Discussion

  • Overall, well written discussion. I wonder what the authors make of their sample having higher rates of exclusive breastfeeding compared to previous reports in their country. It seems from the demographic table that the sample was fairly educated. I wonder how this compares to a national sample and if this may be a factor as well.
  • I agree with the author’s assessment and review of previous literature with regards to parenting stress and depression. As noted above, parents with pre-pregnancy diagnosis of depression/anxiety may also have increased rates of parenting stress. It does not seem as this information was available and should be noted as a limitation.
  • Another potential possibility is how quickly the females are going back to work (if they were employed) and if this is associated with rates of stress. While the authors examined employment status, how quickly parents return to work may also be a factor to consider and/or mention.

Tables/Figures

  • Well organized

Author Response

Reviewer ï¼’

We thank the Reviewer #2 for the valuable comments which we believe would improve the quality of our manuscript. We respond to the comments from the reviewer 1 below one by one.

#1 One key recommendation I have is for authors to clarify what is meant by “healthy females”. Specifically, I am curious if any of the females are known to have depression, anxiety, or other mental health diagnoses. Mild to moderate mental health may be treated without medications and the same can be true for several other chronic disease (e.g., prediabetes may be treated initially through lifestyle change). I am assuming that a female not being on medication is a proxy for having no chronic medical problems but this should be explicit.

→When we recruited our study subjects, as we clarified in exclusion criteria, we only invited mothers who did not have underlying disease that required periodical hospital visit. In other word, our subjects might include woman who have underlying disease but does not require medication. For example, if a mother has subclinical hypothyroidim, the person may be included in our study because she does not need medication (usually hypothyroidim is euthyroid, a condition in which serum levels of TSH <10 μIU/mL and does not require thyroid replacement therapy). Hence, “healthy mothers” in our study subjects might have underlying disease but not require treatment. In this regard, we defined “healthy mothers” as follows.

Line 68-77

  1. Materials and Methods

2.1. Study design, setting and participants

This study was cross-sectional. We recruited Japanese lactating women and their infants aged two and six months after delivery at 73 medical institutions, including 16 hospitals. The remaining sites were obstetrics clinics across all prefectures in Japan. The enrollment period spanned between October 2014 and May 2019. The details of our study were described elsewhere [16]. The inclusion criteria were as follows: 1) healthy singleton infants and 2) healthy mothers who were free from any underlying illness that required periodic hospital visits. In this regard, we defined “healthy mothers” as mothers who do not have underlying illnesses that require ongoing medication.

#2 In addition, while the findings are interesting and I agree with the suggested recommendation that parenting stress should be evaluated, previous data shows high correlation between depression, anxiety and parenting stress. These may be confounding factors which does not appear to have been measured.

→We agreed with the reviewer that our result might be confounded by mental health status of participants. Accordingly, we included the following underlined sentences in the study limitation.

Line 295-311

  1. Limitations

This study had several limitations that need to be addressed. ………Fourth, although our study population consisted of healthy mothers that did not require periodic hospital visits and medication, mild levels of underlying diseases including mental illness or mental health issues during the pre-pregnancy period might influence breastfeeding practice. In addition, we did not measure depressive symptoms that may influence mothers’ perception of parenting stress. Hence, our results require careful interpretation.

#3Introduction

The authors cite some reasons in Line 51-55 on the reasons why some mothers do not exclusively breast feed. While data does support these demographic variables associated with unexclusive breastfeeding, there are other socioeconomic reasons along with the widespread availability of formula that should be considered being mentioned based on applicability in their population. As noted above, it is not clear what is being defined as a healthy mother. Did the authors confirm that mothers did not have any chronic medical diagnoses? What about medical diagnoses that may have milder forms that do not require medication?

→We agreed that our subjects of healthy mothers are defined as women who does not have underlying disease that required ongoing medication. In other word, our study population of mothers might have underlying diseases but do not require ongoing medication. In this regard, we included study limitation that our subjects might have mild form of underlying disease which might have influenced breastfeeding type.

Line 295-311

  1. Limitations

This study had several limitations that need to be addressed. ………Fourth, although our study population consisted of healthy mothers that did not require periodic hospital visits and medication, mild levels of underlying diseases including mental illness or mental health issues during the pre-pregnancy period might influence breastfeeding practice.

#4Methods

Where the rates of maternal depression/anxiety evaluated at all? As noted above, this has been shown to be independently associated with higher parenting stress.

→We did not measure mental health status of participants other than parenting stress. Although our study population is limited to healthy mothers, as the reviewer is concerned, unmeasurement of mental health status may lead to confounding bias. In this regard, we added the following sentences in study limitation section.

Line 295-311

  1. Limitations

This study had several limitations that need to be addressed. ………Fourth, although our study population consisted of healthy mothers that did not require periodic hospital visits and medication, mild levels of underlying diseases including mental illness or mental health issues during the pre-pregnancy period might influence breastfeeding practice. In addition, we did not measure depressive symptoms that may influence mothers’ perception of parenting stress. Hence, our results require careful interpretation.

#5 I think it is fine to limit to Japanese population but I do not know if I would agree with the phrase “race is a fundamental determinant of breastfeeding practices”. I would justify or modify the sentence

→We thank the reviewer for the suggestion to justify the sentence “race is a fundamental determinant of breastfeeding practices” because we were not able to investigate race. Accordingly, we deleted the following sentence.

Line 82-83 in the previous version of manuscript

We limited participation to people of Japanese ethnicity because we hypothesized that race is a fundamental determinant of breastfeeding practices.

#6 Line 91 – What is the meaning of “by post”? This is unclear.

→We changed it to “By postal mail”.

#7 Line 107 – the sentence is written in future tense while the remaining methods is written in past tense. Would maintain consistency

→We changed it to past tense.

Line 105-107

Maternal child-rearing-related stress was assessed using a mother’s child care stress scale (CSS) [17], consisting of a checklist involving three subscales for mental and physical fatigue, worry over child-rearing, and lack of a husband’s support.

#8 It would be helpful to have some brief information about the scale that was used to measure stress. It is not a well-known scale and there is limited data from what I see from a quick literature search

→We agree with the reviewer since the scale we used in our study was developed with Japanese. We included the following explanation why we used the CSS in our study.

Line 109-128

2.4. Data analysis

We used CSS because there were very few scales to measure parenting stress available to assess mothers’ perception of stress in child-rearing nor translated in Japanese. The CSS scale was developed with Japanese and verified to have good reliability and validity [17]. The scale development was based on the psychological stress theory of Lazarus & Folkman [18], and the scale asks mothers how they perceive their parenting environment. By using CSS to clarify mothers’ perceptions of stressful events associated with child-rearing, we were able to examine mothers’ stress and how to intervene in the environment, including related situations and people. In particular, in Japan, there have been many cases of mothers who cannot cope with the stress of child-rearing. This inability leads to a loss of confidence about relating to their children and parents' psychological distress. It may be associated with aggression on behavioral and emotional aspects (i.e., abuse) [19]. The original CSS consists of 33 items on a 9-subscale scale, which was challenging to use in a field study because the follow-up rate may decrease. Instead, we used the short version of the CSS. In this study, we examined the reliability of the short version of the CSS by performing principal factor analyses with Varimax rotation. We confirmed that each item within each factor had high loading (see Appendix). Three domains consisted of childcare exhaustion (Cronbach’s alpha was 0.836 for two months and 0.853 for six months), worry about child development (Cronbach’s alpha was 0.863 for two months and 0.880 for six months), and no support from a partner (Cronbach’s alpha was 0.763 for two months and 0.828 for six months). The factor loadings ranged from 0.547 to 0.878.

#9 Discussion

I wonder what the authors make of their sample having higher rates of exclusive breastfeeding compared to previous reports in their country. It seems from the demographic table that the sample was fairly educated. I wonder how this compares to a national sample and if this may be a factor as well.

→We somehow agree with the reviewer that the reason why our sample has such high prevalence of exclusive breastfeeding practice may be influenced by higher education background of our study participants. However, this hypothesis may be difficult to verify. According to a survey conducted by the Ministry, College and University Enrollment Rate is estimated as 48% in 2003 (https://www8.cao.go.jp/shoushi/shoushika/whitepaper/measures/w-2004/html_h/html/g1221010.html), but the rate does not seem to include graduate degree or higher attainment as we did in our study. Hence in manuscript, we explained with the modest expression as follows (see underlined sentence).

Line 224-228, The first paragraph of Discussion,

  1. Discussion

This study investigated the relationship between parental stress and breastfeeding type among healthy Japanese mothers. The percentage of exclusive breastfeeding exceeded 70%, higher than the Japanese average of 51% [7]. The high prevalence of exclusive breastfeeding in our study may be due to their characteristics of being conscious about breastfeeding in addition to their health status [20].

#10 I agree with the author’s assessment and review of previous literature with regards to parenting stress and depression. As noted above, parents with pre-pregnancy diagnosis of depression/anxiety may also have increased rates of parenting stress. It does not seem as this information was available and should be noted as a limitation.

→We added the following sentence as a study limitation referring to the suggestion from the reviewer.

Line 295-311

  1. Limitations

This study had several limitations that need to be addressed. ………Fourth, although our study population consisted of healthy mothers that did not require periodic hospital visits and medication, mild levels of underlying diseases including mental illness or mental health issues during the pre-pregnancy period might influence breastfeeding practice. In addition, we did not measure depressive symptoms that may influence mothers’ perception of parenting stress. Hence, our results require careful interpretation.

#11 Another potential possibility is how quickly the females are going back to work (if they were employed) and if this is associated with rates of stress. While the authors examined employment status, how quickly parents return to work may also be a factor to consider and/or mention.

→We thank the reviewer for the interesting point to discuss. In Japan, every woman who is in labour must take maternal leave for at least 8 weeks after birth. Hence our participants at two months were supposed to be under the maternal leave (under no pressure how quickly to go back to work). But at 6 months, as the reviewer suggested, mothers might have been under pressure to go back to work. However, our finding that maternal stress was not associated with breastfeeding type at 6 months. Hence, we are not sure if this should be mentioned in text.  
